# Targeting Proliferating Tumor-Infiltrating Macrophages Facilitates Spatial Redistribution of CD8^+^ T Cells in Pancreatic Cancer

**DOI:** 10.3390/cancers14061474

**Published:** 2022-03-14

**Authors:** Xiaobao Yang, Jinrong Lin, Guanzheng Wang, Dakang Xu

**Affiliations:** 1Department of Laboratory Medicine, Ruijin Hospital, Shanghai Jiao Tong University School of Medicine, Shanghai 200025, China; yangxiaobao@sjtu.edu.cn (X.Y.); wanggzzz@163.com (G.W.); 2Department of Sports Medicine, Huashan Hospital, Fudan University, Shanghai 200040, China; linjinrong_fudan@163.com

**Keywords:** tumor microenvironment, tumor-associated macrophages, multiplexed immunohistochemical staining, macrophages depletion, T cell spatial redistribution

## Abstract

**Simple Summary:**

Current studies aim to target tumor-associated macrophages (TAMs) to alleviate immunosuppression and inhibit tumor growth. However, the specific mechanisms that drive TAMs to impede T cell migration into and within tumors are still unclear. Here, we found that after pancreatic ductal adenocarcinoma (PDAC)-bearing mice were treated with clodronate liposomes, the numbers of BrdU-incorporated and Ki-67^+^-proliferating macrophages were reduced, which might be maintained by local proliferation. Clodronate liposomes treatment could alter the macrophages that foster CD8^+^ T cell infiltration, promote CD8^+^ T cell spatial redistribution in tumors, and suppress PDAC growth. This study suggests that depletion of macrophages may increase CD8^+^ T cell infiltration and promote CD8^+^ T cell spatial redistribution in tumors, contributing to the antitumor effect.

**Abstract:**

Tumor-associated macrophages (TAMs) play crucial roles in cancer progression, but the contributions and regulation of different macrophage subpopulations remain unclear. Here, we report a high level of TAM infiltration in human and mouse pancreatic ductal adenocarcinoma (PDAC) models and that the targeting of proliferating F4/80^+^ macrophages facilitated cytotoxic CD8^+^ T-cell-dependent antitumor immune responses. A well-defined KPC-derived PDAC cell line and the murine Panc02 PDAC cell line were used. Treatment of PDAC-bearing mice with clodronate liposomes, an agent that chemically depletes macrophages, did not impact macrophage subpopulations in the local tumor microenvironment (TME). However, further investigation using both BrdU and Ki67 to evaluate proliferating cells showed that clodronate liposomes treatment reduced proliferating macrophages in the KPC and Panc02 models. We further evaluated the distance between CD8^+^ T cells and PanCK^+^ tumor cells, and clodronate liposomes treatment significantly increased the number of CD8^+^ T cells in close proximity (<30 µm) to PanCK^+^ PDAC cells, with increased numbers of tumor-infiltrating IFN-γ^+^CD8^+^ T cells. This study suggests that targeting proliferating tumor-infiltrating macrophages may increase CD8^+^ cytotoxic lymphocyte (CTL) infiltration and facilitate the spatial redistribution of CD8^+^ T cells in tumors, contributing to the antitumor effect.

## 1. Introduction

Clinical and experimental evidence indicates that tumor-associated macrophages (TAMs) are key players in cancers that influence tumor initiation, progression, and metastasis through the complex interplay between tumor cells and their surrounding immune microenvironment [1]. Most evidence indicates that an increased number of TAMs correlates with a poor prognosis [2,3]; however, in some cases, specific TAM subpopulations have been associated with beneficial outcomes [4]. Macrophages have been shown to have their own intrinsic tumoricidal activity and simultaneously promote the activation of cytotoxic lymphocytes. However, they can also show another phenotype to support tumor growth and rapidly adopt immunosuppression in tumors [5]. In some cases, they play a trophic role in tumor development, such as in the embryonic stage [6]. However, the mechanisms that promote TAMs to impede T cell migration into and within tumors remain poorly understood.

Macrophages are found in most tissues after birth and promote the development and remodeling of various organs [7]. The latest progress in gene fate mapping technology has shown that macrophages reside in most tissues in a stable state, develop from embryonic precursors, and are maintained through local proliferation, which may not require input from hematopoietic stem cells (HSCs) of the bone marrow [8]. Recently, Lavine et al. reported that both embryo-derived, tissue-resident macrophages (TRMs) and monocyte-derived TAMs contributed to tumor growth in a pancreatic cancer mouse model [9].

TAMs serve as an ideal target for the treatment of cancer [10]. Although TAM-depletion strategies to treat different tumors have been promoted, the overall benefit for cancer patients was negligible, as reported by extensive clinical studies [11]. The limited success of this approach is due to the plasticity of TAMs. In the tumor microenvironment (TME), TAMs can function as an antitumor component or as a tumor-promoting component. The removal of antitumor TAMs will blunt the efficacy of TAM-depletion therapy [11]. By re-educating TAMs into a proinflammatory and antitumorigenic functional state for treatment, exploiting the tumor-homing ability and plasticity of TAMs can produce effective and lasting responses in different types of cancer, including pancreatic ductal adenocarcinoma (PDAC) [12,13,14]. In PDAC, the density and activation state of infiltrating macrophages are related to disease progression and treatment resistance [15,16]. Recent evidence has indicated that TAMs are an important factor that influences the total number of T cells, as well as the T cell distribution and the ability of T cells to enter tumor islets in lung cancer [17]. However, the mechanisms by which TAMs prevent CD8^+^ T cells from the periphery from reaching tumor cells are still unknown. Moreover, current knowledge on the interaction among macrophages, CD8^+^ T cells, and PDAC is still limited, and the effect of TAMs on local environmental factors that control the motility of T cells needs to be investigated.

Based on the complex heterogeneity of macrophages, the role of TAMs in the TME is crucial in current TAM-targeting strategies. Preclinical and clinical data indicate that a high degree of TAM infiltration is related to the poor prognosis of some types of cancer [18], such as glioblastoma [19] and bladder cancer [20]. On the other hand, in some cancers, such as colorectal cancer [21] and ovarian cancer [22], TAM infiltration is associated with a good prognosis. The difference between these results can be attributed to not only different cancer types but also some intratumor factors, such as the TAM distributions in the different TMEs. Elevated levels of TAMs in the tumor stroma are associated with poor prognosis in non-small-cell lung cancer (NSCLC), and the degree to which TAM penetrate cancerous islets is associated with good prognosis [23,24]. These findings demonstrate the inter- and intratumoral heterogeneity of TAMs, which may be related to the ontogeny and location of TAMs in the TME. The activation status of TAMs, such as M1 or M2 polarization, also contributes to cancer progression; for example, in PDAC, M2-polarized TAMs, rather than general TAMs, are correlated with metastasis and poorer prognosis in PDAC patients [3,25]. To distinguish the different roles of TAMs under various conditions, the need to redefine TAM subsets and their functions in the TME by integrating new technologies, such as multiplexed immunohistochemistry (mIHC), mass cytometry by time-of-flight (CyTOF), single-cell RNA-seq (scRNA-seq), spatial transcriptomics, and systems biology approaches, is urgent [26]. Once we redefine the subsets of TAMs, we can properly target these myeloid-derived, tumor-infiltrating macrophages, halt PDAC tumor progression, and improve antitumor immunity and clinical outcomes. A growing body of published reports on TAMs in PDAC have comprehensively reviewed the origin, polarization, roles, and reprogramming of TAMs in pancreatic cancer [27]; the re-education of TAMs; and the therapeutic implications associated with targeting TAMs in pancreatic cancer—all of which are related to defining TAM subsets. Chemokine pathways are co-opted in PDAC to facilitate TAMs from the bone marrow to establish an immunosuppressive TME [28]. As another example, TAMs promote PDAC progression by modulating the Warburg effect through the CCL18/NF-κB/VCAM-1 pathway [29]. This network might provide a potential new therapeutic strategy for PDAC. The ideal approach would be to define the subset of TAMs that can contribute to PDAC development and to carry out therapeutic intervention.

Here, we investigated the impact of targeting TAMs by using clodronate liposomes, an agent for the chemical depletion of macrophages, to treat Panc02 and KPC tumor models and explored TAM complexity. The results demonstrated that targeting proliferating F4/80^+^ macrophages might alter the myeloid population and is associated with reduced Foxp3^+^ Tregs and increased CD8^+^ T cell infiltration. Targeting proliferating macrophages also enhanced the antitumor effect of CD8^+^ T lymphocytes and promoted CD8^+^ T cell spatial redistribution in tumors, contributing to a more antitumor immune environment.

## 2. Materials and Methods

### 2.1. Patient Samples and Mouse Models

All human PDAC tissues were obtained from Ruijin Hospital, Shanghai Jiao Tong University School of Medicine. The protocols using human specimens were approved by the Ethics Committee of Ruijin Hospital, Shanghai Jiao Tong University School of Medicine (ID: 2021-194).

For the studies involving animals, appropriate permission was given from the Animal Experimental Ethical Inspection of Ruijin Hospital, Shanghai Jiao Tong University School of Medicine (20180509). A sample of 6-week-old female C57BL/6 mice were injected subcutaneously with 5 × 10^5^ Panc02 cells or 3 × 10^5^ KPC cells (mouse pancreatic cancer cell lines) on day 0. For macrophage-depletion studies, mice were intraperitoneally injected with clodronate liposomes or control liposomes (FormuMax Scientific, Sunnyvale, CA, USA) at 200 μL per mouse on days -1 and 0, and every 3 days afterward until the completion of the study. In the T-cell-depletion experiment, an antibody (Ab) to deplete CD8^+^ cells (clone YTS 169.4; BioXCell) was injected intraperitoneally at 200 µg per mouse on days -1 and 0, and every 5 days afterward until the end of the study. The tumor volume was calculated as width^2^ × length/2. On day 20, all mice were sacrificed, and tumor tissues were collected for further analysis.

### 2.2. Cell Culture

Two different mouse PDAC cell lines were used: Panc02 cells are a chemically induced cell line [30], and KPC cells were derived from tumors in situ with *KRAS* and *p53* mutations [31]. Both cell lines were a kind gift from Dr. Clare Slaney (Peter MacCallum Cancer Centre, Melbourne, Australia) and were cultured in Dulbecco’s modified Eagle medium (DMEM; HyClone, Logan, UT, USA) supplemented with 10% fetal bovine serum (FBS; Gibco, Invitrogen, Carlsbad, CA, USA), streptomycin (100 μg/mL), and penicillin (100 U/mL). Mycoplasma contamination testing (PlasmoTest Mycoplasma Detection Kit, InvivoGen, San Diego, CA, USA) was performed to confirm that no mycoplasma contamination was present. Cells were cultured at 37 °C in an atmosphere containing 5% CO_2_. For coculture analysis, TAMs and the Panc02 or KPC cell line were cocultured using a noncontact coculture transwell system (Corning, NY, USA). TAMs isolated from tumor tissues were seeded in 0.4 μm pores (1 × 10^5^ cells per pore), and 6-well plates were seeded with Panc02 or KPC cancer cells (1 × 10^5^ cells per well). We administered clodronate liposomes or control liposomes to the cancer cells. After 48 h of coculture, the Panc02 and KPC cells were harvested for further analyses.

### 2.3. Bioinformatic Analysis of Immune Cell Subtypes in PDAC

The RNA sequencing (RNA-seq) data of PDAC patients were obtained from the TCGA database. The TCGA-PAAD datasets constituted 178 tumor and 4 normal samples. All tumor samples were utilized for further analysis. The CIBERSORT algorithm is a deconvolution algorithm that calculates the cell composition of tissues based on gene expression profiles [32]. To explore the tumor immune landscape in the PDAC samples, we normalized RNA-seq gene expression in the TCGA-PAAD dataset, uploaded to the CIBERSORT website (https://cibersort.stanford.edu/index.php (accessed on 5 October 2020)), and then computed the data by the LM22 gene signature (a “signature matrix” of 547 genes) to characterize the immune cell composition [32].

GEPIA is an interactive website that includes data from TCGA and the GTEx projects [33]. The GEPIA website was accessed to analyze the expression of CD68, CD86, and CD163 in pancreatic cancer tissues (*n* = 179) and normal tissues (*n* = 171); log_2_(TPM + 1) transformed expression data were chosen for plotting.

For overall survival analysis based on CD68, CD86, and CD163 expression, 179 patients were divided into either a high-expression group or a low-expression group using the quartile value of TPM expression as the cutoff value based on the GEPIA data. The log-rank test was used for the analysis of overall survival.

### 2.4. Immunohistochemical Analysis

Paraffin slides were used for immunohistochemistry (IHC) for F4/80 (a marker of macrophages). First, the paraffin slides were baked at 60 °C overnight and then deparaffinized by xylene and rehydrated with ethanol. Sodium citrate antigen-retrieval solution (Beyotime, Shanghai, China) was used for retrieval with a low-level microwave. Hydrogen peroxide (1.5%) in methanol was used to block endogenous peroxidase activity, and Dako protein-blocking reagent (DakoCytomation Corporation, Carpinteria, CA, USA) was used to block the irrelevant antigen. The primary Ab F4/80 (D2S9R, working concentration of 1:500, Cell Signaling Technology, Danvers, MA, USA) was incubated overnight at 4 °C. For isotype control slides, normal rabbit IgG (Calbiochem, San Diego, CA, USA) was used as the primary Ab. A Dako Real Detection System Alkaline Phosphatase Kit (K5005, Dako) and chromogen (Fast Red) were used for immunostaining. Hematoxylin was used to counterstain, and the slides were then dehydrated.

### 2.5. Multiplexed IHC and Immunofluorescence (IF) Analysis

Formalin-fixed pancreatic tissue from human tissue specimens and in vivo experiments in mice were subjected to multiplexed IHC (mIHC) and IF analysis. The detailed staining protocol has been described in our previous publications [34]. The Abs used were purchased from Cell Signaling Technology: anti-CD68 (D4B9C), anti-CD86 (E2G8P), anti-CD163 (D6U1J), anti-pan-keratin (C11), anti-F4/80 (D2S9R), anti-Ki-67 (D3B5), and anti-CD8 (D4W2Z); an Opal 4-color Manual IHC Kit (PerkinElmer, Spokane, WA, USA) was used for mIHC. The Vectra Automated Quantitative Pathology Imaging System (PerkinElmer) was used to capture the mIHC staining results. The detailed protocol has been described in our previous studies [35]. InForm software (version 2.4.8, Akoya Biosciences, Menlo Park, CA, USA) was used to analyze the phenotype and spatial information. To quantify the expression levels of F4/80, Ki-67, CD8, and Foxp3 in tumors, 10 fields were randomly selected for assessment. The cells of interest in each section were counted, and this value was then divided by the total number of cells which mean DAPI-positive cells to calculate the percentage of positive cells. Quantitative imaging analysis of data exported from InForm software was performed using the “R” packages “Phenoptr” and “PhenoptrReports” to analyze the distance information and prepare images.

### 2.6. Bromodeoxyuridine (BrdU) Labeling

A duration of 24 h prior to euthanasia, mice were injected intraperitoneally with BrdU (100 µg/g body weight, Sigma, St. Louis, MO, USA) every 8 h. Each mouse received a total of 3 injections. Eight hours after the final injection, the mice were sacrificed, and BrdU incorporation in macrophages was analyzed by flow cytometry.

### 2.7. Flow Cytometric Analysis

Half of each tumor was digested with enzymes and processed to generate single-cell suspensions, as described previously [36]. Fresh tumor tissues were cut into small pieces and subsequently digested with a collagenase IV (300 U/mL)/hyaluronidase solution (200 U/mL) in RPMI 1640 medium, supplemented with penicillin and streptomycin at 37 °C for 1–2 h. The cell suspensions were then centrifuged at 350× *g* for 5 min at 4 °C, followed by macrophage isolation, Ab labeling, and flow cytometric analysis. The Abs used for flow cytometry were purchased from BioLegend (San Diego, CA, USA). For macrophage analysis, anti-CD45-FITC (clone 30-F11), anti-CD11b-PerCP-Cy5.5 (clone M1/70), anti-F4/80-APC (clone BM8), anti-CD68-PE (clone FA-11), anti-MHC-II-PE (clone M5/114.15.2), anti-CD86-PE (clone GL-1), anti-CD163-PE (clone S15049I), anti-CD206-PE (clone C068C2), anti-CX3CR1-PE (clone SA011F11), and anti-CCR2-PE (clone SA203G11) were used for different panels. For T cell analysis, anti-CD45-FITC (clone 30-F11), anti-CD3-PECy7 (17A2), anti-CD4-APC (clone GK1.5), anti-CD8-PerCP-Cy5.5 (clone 53-6.7), and anti-Foxp3-PE (clone MF-14) were used. The True-Nuclear^TM^ Transcription Factor Buffer Set (BioLegend, San Diego, CA, USA) was used for Foxp3 staining. Dead cells were excluded using the Zombie NIR Fixable Viability Kit (BioLegend). To identify apoptotic and necrotic tumor cells with or without coculture with macrophages isolated from murine tumor tissues using the mouse F4/80^+^ Isolation Kit (Miltenyi Biotec, Auburn, CA, USA), a BioLegend FITC Annexin V apoptosis detection kit with propidium iodide (PI) was used. For the BrdU assay, BrdU incorporation was detected by the FITC BrdU Staining Kit for Flow Cytometry (Invitrogen). Data acquisition was performed on an LSRII system (BD Biosciences), and FlowJo software (version V10, Becton Dickinson, Ashland, OR, USA) was used for analysis.

### 2.8. Statistical Analysis

Statistical differences in tumor kinetics between the 2 different treatment groups were determined using Student’s t test, and one-way ANOVA was performed for multiple comparisons. To assess the difference in the number of immune cells in tumors by flow cytometry or mIHC, statistical significance was evaluated using Student’s *t* test or one-way ANOVA. A *p* value < 0.05 was considered to indicate significance (*: *p* ≤ 0.05, **: *p* ≤ 0.01, ***: *p* ≤ 0.001). Graphs were made using GraphPad version 8.0 software (GraphPad, La Jolla, CA, USA).

## 3. Results

### 3.1. High Level of TAM Infiltration in Human and Mouse PDAC

The TME of human PDAC is characterized by the infiltration of a large number of immune cells, of which the most important infiltrating immune cells are macrophages [37]. To further understand the presence of macrophages in the immune microenvironment, the CIBERSORT algorithm was used to analyze the proportions of tumor-infiltrating immune subgroups, and 22 immune cell profiles were constructed using PDAC samples (Figure 1A). Among them, the levels of four kinds of tumor-infiltrating immune cells—M0 macrophages, M2 macrophages, CD4^+^ T cells, and CD8^+^ T cells—were elevated in humans, with macrophages the most abundant population in the infiltrated immune cells. These results further supported the conclusion that the immune activity of the TME was highly affected. To further assess the abundant macrophages infiltration in the TME, we performed further analysis of the expression of the macrophage markers CD68, CD86, and CD163. These individual markers were significantly overexpressed in PDAC tissues compared with adjacent normal tissues, and the pan-macrophage marker CD68 was significantly associated with the overall survival of PDAC patients (Figure 1B and Appendix A). This could also be readily observed through comparison of the number of cells that expressed the macrophage markers CD68, CD86, and CD163 in paired human tumor lesions and adjacent normal tissue in the same tissue block (Figure 1C). Other representative images of increased macrophage markers in the tumor lesions compared to the patient’s own adjacent normal area are shown in Appendix A. Parallel to the human data, an increase in the macrophage number was also found in our pancreatic cancer mouse models (Panc02 and KPC cells) compared with spleen tissues (Figure 1D). Moreover, IF staining also identified a significant portion of Ki-67^+^F4/80^+^ cells in the high Ki-67-stained tumor area of Panc02 or KPC cells (Figure 1E). These results indicated a high level of TAM infiltration into human patient tumors and mouse PDAC and that in situ proliferation correlated with Ki-67^+^F4/80 density. Although infiltrating macrophages have characteristic manifestations in tumors, the function of these cells is not well characterized.

### 3.2. Targeting F4/80^+^ Macrophages Impaired Tumor Progression in Panc02 and KPC Pancreatic Cancer Models

To determine whether macrophages regulate tumor progression, we targeted macrophages in mice by administering clodronate liposomes prior to the subcutaneous injection of 5 × 10^5^ Panc02 cells or 3 × 10^5^ KPC cells and throughout the experiment (Figure 2A). Clearly decreased numbers of macrophages (CD11b^+^F4/80^+^) and without pronounced depletion of CD11b^+^ population were found in the spleens after intraperitoneal injection of clodronate liposomes (Figure 2B and Appendix A). The clodronate liposomes treatment was also found to inhibit tumor growth and decrease tumor weight (Figure 2C,D). The above in vivo study showed that administering clodronate liposomes inhibited tumor progression in murine models of PDAC. Next, we wanted to investigate the modulation of tumor suppression through immune or tumor components in the TME. We administered clodronate liposomes or control liposomes to the Panc02 and KPC cells to determine whether clodronate liposomes would affect the survival of cancer cells directly. As shown in Figure 2E, when we directly treated Panc02 and KPC cells, or cocultured Panc02 or KPC cells with murine macrophages, which were isolated from the tumor tissues with anti-F4/80 microbeads, the application of clodronate liposomes did not alter cellular necrosis or apoptosis compared with those of control cells (Figure 2E). These findings confirm that macrophages indeed contribute to tumor progression. However, which subpopulation of macrophages contributes to PDAC progression is still unknown.

### 3.3. Macrophage Subsets and Functional Analysis of the TAM-Targeting Strategy

Macrophages exhibit an enormous amount of plasticity: they exert proinflammatory/antitumor/M1 properties or anti-inflammatory/tumor-promoting/M2 properties, depending on stimulation of the local microenvironment. Since Panc02 and KPC tumor lesions contain considerable numbers of recruited macrophages [38,39], the remainder of the study focused exclusively on characterizing macrophage subpopulations in tumor lesions. Immunohistochemical analysis of tumor sections for F4/80^+^ macrophages showed that F4/80^+^ macrophage numbers increased as tumor lesions progressed to palpable tumors (Figure 3A), suggesting that these cells may have the capacity to promote tumors; however, there was no alteration in the proportion of F4/80^+^ macrophages upon the administration of clodronate liposomes compared to that with control treatment (Figure 3B). Next, we investigated the subsets of macrophages. Flow cytometric analysis of tumor lesions with or without clodronate liposomes treatment was performed to define the macrophage subtypes. The CD11b^+^ population remained relatively unchanged (Appendix A). After gating the live CD45^+^ CD11b^+^ subpopulation, we evaluated expression of the following cell surface markers on F4/80^+^ cells: the pan-macrophage marker CD68, the proinflammatory macrophage markers MHC-II and CD86, and the anti-inflammatory macrophage markers CD163 and CD206 and the gating strategies were shown in Appendix A. Treatment of Panc02 tumors with clodronate reagents strongly reduced the F4/80^+^ cell population in the spleen (Figure 2B). However, numbers of the F4/80^+^ cell population and its subpopulation remained unchanged after targeting macrophages that were characterized by flow cytometry analysis from the tumor (Figure 3C–G). In addition, the identification of monocyte-derived TAMs and TRMs in the TME was necessary. Inflammatory monocytes/macrophages have been shown to express both CC chemokine receptor 2 (CCR2) and CX3C chemokine receptor 1 (CX3CR1). Regardless of whether clodronate liposomes were administered, we found a large population that consisted of 20–70% CX3CR1^+^F4/80^+^ cells, which may indicate some inference by TRMs, and a 5–15% CCR2^+^F4/80^+^ population was found in the tumor. CCR2 is the receptor of CCL2, and TAMs that express CCR2 play a key role after being recruited to most tumors (Figure 3F). Together, the above results indicate that infiltrating macrophages are dynamic in the process of tumor lesion progression and can express a spectrum of proinflammatory and anti-inflammatory markers. Importantly, because TAMs are upregulated in tumor tissues, replenishing a portion of the TAM population by recruiting a portion of the circulating monocytes may lead to an unaffected F4/80^+^ subset of macrophages in local tumor lesions.

### 3.4. Impact of Inhibiting F4/80^+^ Macrophage Proliferation on Tumor Progression in Panc02 and KPC Tumor Models

Considering macrophage function and to examine whether TRMs expand through local proliferation given the number of unaffected F4/80^+^ macrophages, we analyzed Ki-67 expression in macrophages. Ki-67 and BrdU are broadly used as proliferation markers and usually show similar expression patterns. The Ki-67 protein is a nuclear antigen associated with cell proliferation and can be used as a marker in cell proliferation assays because it is expressed throughout the active cell cycle (G1, S, G2, and M phases) but not in the resting phase (G0). IF staining identified a significant proportion of Ki-67^+^F4/80^+^ cells in Panc02 or KPC PDAC tissues with or without clodronate liposomes treatment, and the proportion of Ki-67^+^F4/80^+^ cells was significantly reduced (Figure 4A,B). The BrdU cell proliferation assay is a versatile and convenient method for the quantification of cell proliferation, as BrdU can be used to label nascent DNA in living cells. To confirm the local proliferation of F4/80^+^ cells, we further analyzed the PDAC tissues, and the results demonstrated that approximately 8% of F4/80^+^ macrophages in the Panc02 model and 6% in the KPC model incorporated BrdU within 24 h. In contrast, the proportions were strongly reduced in the clodronate-liposomes-treated group (Figure 4C). In addition, macrophages treated with clodronate liposomes exhibited a decreased number of Foxp3^+^ Tregs (Figure 4D) and an increased number of CD8^+^ T cells (Figure 4E). These findings may be related to the inhibition of tumor growth.

### 3.5. Targeting Proliferating F4/80^+^ TAMs Improved CD8^+^ T Cell Infiltration

To determine whether the efficacy of targeting proliferating F4/80^+^ macrophages was CD8^+^ T-cell-dependent, tumor-bearing mice were administered anti-CD8 depleting Abs during the course with or without clodronate liposomes treatment. CD8^+^ T cell depletion was confirmed by flow cytometric analysis of splenocytes collected from the mice at the study endpoint (Figure 5A). Administration of anti-CD8 depleting Abs abrogated the efficacy of impairing proliferating F4/80^+^ macrophages in mice bearing Panc02 tumors (Figure 5B) and KPC tumors (Figure 5C). Next, targeting the tumor-related F4/80^+^ macrophages improved the infiltration of CD8^+^ T cells in the tumors. In Panc02 and KPC tumor models, targeting TAMs significantly increased the infiltration of CD8^+^ T cells (Figure 5D). Upon comparison of the treated and control groups, the total number of infiltrated CD45^+^ hematopoietic cells remained unchanged. In Panc02 and KPC tumors, the total CD4^+^ T cell populations remained relatively similar. However, CD4^+^Foxp3^+^ Treg numbers were reduced in the TAM-targeted group (Figure 5E). There were more Foxp3^+^ Tregs in the Panc02 model, which is consistent with previous studies [40]. These data support the ability of targeting proliferating F4/80^+^ macrophages to enhance a CD8^+^ T-cell-dependent antitumor effect in vivo.

### 3.6. Targeting Proliferating F4/80^+^ TAMs Promoted CD8^+^ T Cell Spatial Redistribution in Tumors

Similarly, increased total CD8^+^ T cell numbers were shown in mice treated with clodronate liposomes (Figure 5D). We further investigated the spatial redistribution of CD8^+^ T cells in Panc02 and KPC tumors after targeting macrophages in mice. To determine whether clodronate liposomes, which induced local environmental changes, could change the proximity of T cells relative to PDAC cells, we double stained for pan-keratin (PanCK) and CD8^+^ CTLs and analyzed cell proximities (Figure 6A–D). It is worth noting that CD8^+^ T cells were mainly located at the edge or periphery of the tumor without treatment, but in the clodronate liposomes treatment group, CD8^+^ T cells were enriched around the tumor core or center (Figure 6A,B). When we further evaluated the distance of CD8^+^ to the tumor cells (PanCK), we found that, in control-treated mice, there were few CD8^+^ T cells at distances less than 30 µm, with an average distance of CD8^+^ T cells to tumor cells of approximately 22 µm for Panc02 cells and 40 µm for KPC cells. In contrast, clodronate liposomes treatment significantly increased the number of CD8^+^ CTLs in close proximity (<30 µm) to PanCK^+^ PDAC cells (Figure 6C,D). Corresponding with this targeting of F4/80^+^ TAMs was not only an increase in the number of CD8^+^ T cells but also the promotion of CD8^+^ T cell spatial redistribution and effector functions in tumors. We observed increased numbers of tumor-infiltrating IFN-γ^+^CD8^+^ T cells in Panc02 and KPC tumors treated with clodronate liposomes (Figure 6E), and the tumor-infiltrating GZMB^+^CD8^+^ T cells were remained relatively unchanged (Appendix A). More recent data indicated that the spatial distribution of tumor-infiltrating CD8^+^ T cells refines their prognostic utility for pancreatic cancer survival [41]. The increase in effector CD8^+^ T cells was consistent with the spatial distribution of tumor-infiltrating CD8^+^ T cells (Figure 6A–D). Our spatial computational analysis of PDAC reveals the functional validity of CD8^+^ cell density in the tumor center. Together, these findings highlight the critical contribution of targeting TAMs in promoting the trafficking of tumor-reactive CD8^+^ T cells and their redistribution.

## 4. Discussion

Targeting TAMs aims to inhibit tumor growth and immunosuppression and avoid inflammatory phenotypes that may be related to antitumor function. However, the specific mechanisms that drive TAMs to impede T cell migration into and within tumors are still unclear. Here, we found unexpected results: when PDAC-bearing mice were treated with clodronate liposomes, we expected the F4/80^+^ macrophage numbers in the spleen and tumor sites to be reduced, but our data showed only a minimal impact on the intratumoral F4/80^+^ macrophage and macrophage subpopulations in the local TME. Further investigation found reduced numbers of proliferating F4/80^+^ macrophages, which might be derived from embryonic precursors and maintained by local proliferation. Clodronate liposomes treatment can lead macrophages to foster CD8^+^ T cell infiltration, promote CD8^+^ T cell spatial redistribution and its effect function in tumors, and suppress PDAC growth.

Between peripheral and local macrophages, M1 macrophages are strongly repolarized to the M2 phenotype. Such repolarization is decisive for tumor progression and may be a key reason for the failure of tumor-directed TAM targeting, as reported here. In our study, when clodronate liposomes were used to treat PDAC-bearing mice, our data showed little effect on the numbers of F4/80^+^ macrophages and macrophage subpopulations within the tumor. We did not find that the obvious subpopulation of TAMs was altered by clodronate liposomes, except for proliferating F4/80^+^ TAMs, including a subtype expressing the macrophage scavenger receptor CD163, which is a prototypical marker of anti-inflammatory macrophages that are negatively associated with prognosis in many types of cancer. CD163 expression is related to increased MHC-II expression and decreased inflammatory cytokine secretion. However, interestingly, clodronate liposomes treatment had a less pronounced antitumor effect in KPC tumors than in Panc02 tumors, which may be due to less CD206 M2 expression and the low number of CX3CR1-expressing TAMs (left side of Figure 3E,F). These results indicate that infiltrating macrophages are not greatly affected by macrophage-depletion strategies in cancer progression, unlike peripheral macrophages and TAMs, which can still exert proinflammatory and anti-inflammatory effects after clodronate liposomes treatment. To determine whether turnover of resident macrophages occurs through replacement by blood monocytes, we evaluated F4/80- and CCR2-expressing macrophages, but these cells were not altered by clodronate liposomes treatment, so the macrophages in the local tumor site were similar to TRMs. This may be one of the key reasons for the failure to effectively suppress tumor growth by directly targeting macrophages, which may be a specific subset of TAMs.

Interestingly, further investigation found that treatment with clodronate liposomes reduced the numbers of proliferating F4/80^+^ macrophages, which are TAMs that may be derived from embryonic precursors and are maintained by proliferation through the local environment. Recent reports indicated that resident macrophages had proliferation capacity in situ and were self-maintained locally in organs, including the lungs, heart, and brain, and in PDAC tumors [9,42,43]. However, studies have yet to determine whether clodronate liposomes can affect resident macrophages that proliferate in situ. This motivated our assessment of the proliferative capacity of resident macrophages in situ in tumors treated with clodronate liposomes. Accordingly, F4/80- and Ki-67-positive macrophages were found closer to tumors in PDAC tissue than F4/80- and Ki-67-negative macrophages were. Macrophage depletion is an approach that can be applied under pathological conditions and is based on the use of depleting Abs (such as anti-CSF-1 receptor (CSF-1R)) or molecules specifically cytotoxic to macrophages (such as clodronate liposomes). In mouse models of cervical cancer and breast cancer, the depletion of TAMs achieved with highly selective CSF-1R inhibitors led to tumor growth arrest or delay [44]. The modulation of CSF-1R signaling was shown to have significant impacts on the number and distribution of macrophages in a study of glioblastoma multiforme (GBM). Interestingly, CSF-1R inhibitors were used to target TAMs in the mouse GBM model. The inhibitor could significantly improve the survival rate and tumor regression, but more than half of tumors are resistant to the treatment [45]. This result suggests that other factors are also related to the survival of macrophages or that the survival of some macrophages does not depend on CSF-1R. We chose clodronate liposomes treatment to specifically target proliferating F4/80^+^ macrophages that may reprogram TAMs to suppress tumor growth. To answer the question of why clodronate liposomes did not deplete the macrophages in tumors, it comes to several reasons. In the PDAC model, (1) using clodronate liposomes to deplete macrophages, we speculated that the loss of TAMs can be compensated by the recruitment of circulating Ly6G^hi^ monocytes. (2) The use of CSF1-neutralizing antibodies combined with clodronate liposomes to deplete TAMs resulted in only a 50% reduction in TAMs in established tumors. (3) When treated with one dose of CSF1-neutralizing antibodies on embryonic day 13.5 (E13.5), an 80% reduction in TRMs was observed at 6 weeks of age [9]. This means that significant portions of TRMs originated from embryonic development and expanded through in situ proliferation during tumor progression. Optimal therapeutic intervention requires an in-depth understanding of the sources that sustain macrophages in malignant tissues. The use of clodronate liposomes to deplete TAMs, at least targeting proliferating F4/80^+^ macrophages, may need to be combined with CCR2-neutralizing antibodies to lock the circulating Ly6G^hi^ monocytes and increase efficacy.

In the TME, infiltrating macrophages are dynamic and exhibit enormous plasticity. At early stages, tumor lesions represent a higher expression of immune stimulatory genes (MHC-II, etc.)s resembling a proinflammatory M1-like phenotype; when progressing to advanced invasive cancer, the expression of immunosuppressive gene (CD206, etc.) is significantly upregulated, becoming the M2-like phenotype [46]. In many types of tumors, TAMs often acquire the characteristics of a polarized M2 phagocytic cell population and play a crucial role in promoting tumor growth and cancer development. Previous studies have shown that anti-CSF-1R therapy alone can change the phenotype of macrophages to the antitumor M1-like subtype in vivo, thereby exerting a powerful antitumor effect. In our study, we expected that the inhibition of tumor growth achieved by clodronate liposomes treatment would be mediated in part by altering the M1/M2 polarization of TAMs, resulting in significant increases in the subpopulation of M1 macrophages and the proportion of M1/M2 macrophages in the TME. However, we did not find that clodronate liposomes regulated the polarization of macrophages to support an M1-like state (Figure 3B,C). We also did not find that the proportion of M1 macrophages and the ratio of M1/M2 macrophages were increased in the tumor tissues of mice injected with Panc02 or KPC cells and treated with clodronate liposomes. An important method of communication between tumor cells and immune cells may involve tumor-derived cytokines and secreted factors. This interaction needs to be confirmed by coculture studies of tumor cells and macrophages. Tumor cells can reprogram the function of immune cells, induce immune cell dysfunction, and suppress the antitumor immune response, thereby suppressing the immune response. We showed that targeting proliferating F4/80^+^ macrophages fostered CD8^+^ T cell infiltration and promoted CD8^+^ T cell spatial redistribution, thereby enhancing antitumor immunity. In general, macrophages can prevent CD8^+^ T cell infiltration in the TME, and since they participate in antigen presentation, targeting macrophages has the benefit of priming the TME for T cell infiltration [47]. Additionally, Yuan et al. reported that reduced macrophages infiltration and high CD8^+^ T cell infiltration into the tumor area could be induced by the abolition of AngII production or AngII signal blockade [48]. Seok et al. reported that Cyclooxygenase-2 (COX-2) may also be a macrophages-related factor that can prevent CD8^+^ T cell infiltration in the TME [49].

Traditionally, PDAC is considered an immunologically silent malignant tumor, but recent treatment strategies have shown that effective immune-mediated tumor cell death can be exploited by PI3K-regulated disease transmission. The basic principle of these strategies is that if immunosuppressive disorders are eliminated, CD8^+^ T cells can be mobilized to recognize and eliminate malignant cells. The data provided in this paper revealed the key role of CD8^+^ T cell functional regulation in PDAC and identified TAMs as a key regulator of tumor immunosuppression. Some subpopulations of TAMs may produce CCL2, CXCL9, and CXCL10, suggesting CD8 T cell trafficking to tumors [50]. The data in Figure 5B further support the notion of antitumor immunity through CD8 T cell recruitment. Some cytokines/chemokines, such as CCL22, CCL28, CXCL12, CCL5, and CCL1, can be produced by macrophages [51]. This leads Tregs to be attracted to the tumor sites, which can increase Treg infiltration and decrease CD8+ cell infiltration [52]. Therefore, it would be interesting to study some of the cytokines/chemokines that are released by the subpopulation of macrophages and attract the migration of Tregs, leading to Treg immunosuppression. Our research showed that relative to those of local tumors, both human and murine PDAC exhibited increased levels of TAMs, but not peripheral blood cells. In the treatment of PDAC at different stages, clodronate liposomes slowed the progression of tumors in a T-cell-dependent manner. At the same time, as tumor growth slowed, the percentage of CD8^+^ T cells with enhanced effector molecule expression significantly increased in pancreatic tumors because the depletion of CD8^+^ T cells eliminated the benefits of macrophage targeting. An increase in the proportion of cells with CD8^+^ T cell phenotypes was also observed during clodronate liposomes treatment, which is consistent with other studies on the response of CD8^+^ T cells to various immunotherapies. Based on these data, macrophage targeting leads to T cell activation and promotes tumor regression. Interestingly, when Panc02 or KPC cells were analyzed based on clodronate liposomes treatment administration and the intratumoral and peritumoral densities of CD8^+^ T cells, a higher probability of response was observed in the mice with high intratumoral but not peritumoral CD8^+^ T cell levels (Figure 6). In the future, which subpopulation of TAMs is affected by clodronate liposomes still needs to be further investigated. In addition, it would be interesting to look for functional cell subsets and populations which exert an immunosuppressive effect in the PDAC microenvironment. Specifically, this would be to detect the following: (1) effective and immunosuppressive phenotypes of infiltrating CD4 T cells to see whether there is a switch from Th1 “effector” cells (IFN-r^hi^, IL-10^low^, PD-1^low^) to cells with a suppressive phenotype (IFN-r^low^, IL-10^high^, PD-1^high^); (2) the functional state of CD8, which might switch from an “effector” phenotype (IFN-r^hi^, CD107^high^, PD-1^low^) to an “exhausted” phenotype (IFN-r^low^, CD107^low^, PD-1^high^); (3) the expression of PD-L1 in resident macrophages; (4) the infiltrating Treg cells with a more immunosuppressive phenotype (IL-10^high^).

## 5. Conclusions

The reduction in Panc02 and KPC tumor growth in mice treated with clodronate liposomes indicates that targeting TAMs is a potentially successful treatment for the management of pancreatic cancer. The above results suggest that macrophage-directed therapy is an important controlling factor in pancreatic cancer, derived through an unexpected mechanism in which the targeting of proliferating F4/80^+^ TAMs improves CD8^+^ T cell infiltration and promotes CD8^+^ T cell spatial redistribution in tumors.

## Figures and Tables

**Figure 1 cancers-14-01474-f001:**
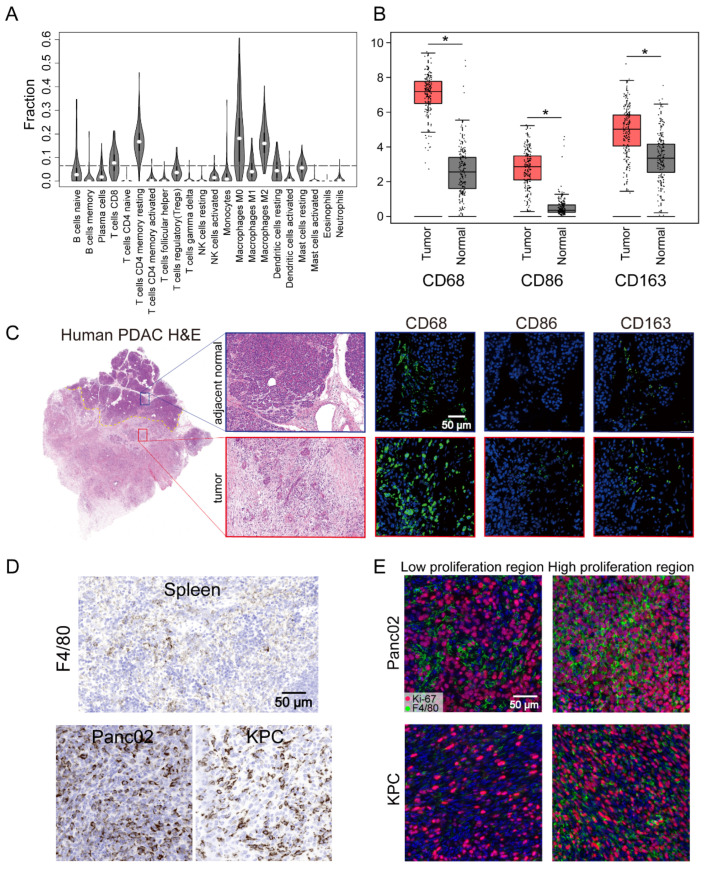
Pancreatic ductal adenocarcinoma (PDAC) tumors were highly infiltrated by macrophages. (**A**) Tumor-infiltrating immune cell abundance profiles determined by using a violin plot showing the ratio differentiation of 22 kinds of immune cells in PDAC tumor samples (*n* = 178) from the TCGA-PAAD datasets. (**B**) The transcriptome data from GEPIA database showed that CD68, CD86, and CD163 were more highly expressed in PDAC tissues (*n* = 179) compared with normal tissues (*n* = 171). (**C**) Representative images of human PDAC and adjacent normal pancreatic tissues assessed for macrophage density (CD68, CD86, or CD163). (**D**) Representative IHC images of pancreatic tissue from the *KRAS* and *p53* mutants (KPC) and Panc02 mouse models showing a high infiltrated macrophage (F4/80) density in comparison to that in images of spleen tissues. (**E**) Representative immunofluorescence images of Ki-67^+^F4/80 staining in Panc02 and KPC tumors. Statistical analyses were carried out by Student’s *t* test. *: *p* ≤ 0.05.

**Figure 2 cancers-14-01474-f002:**
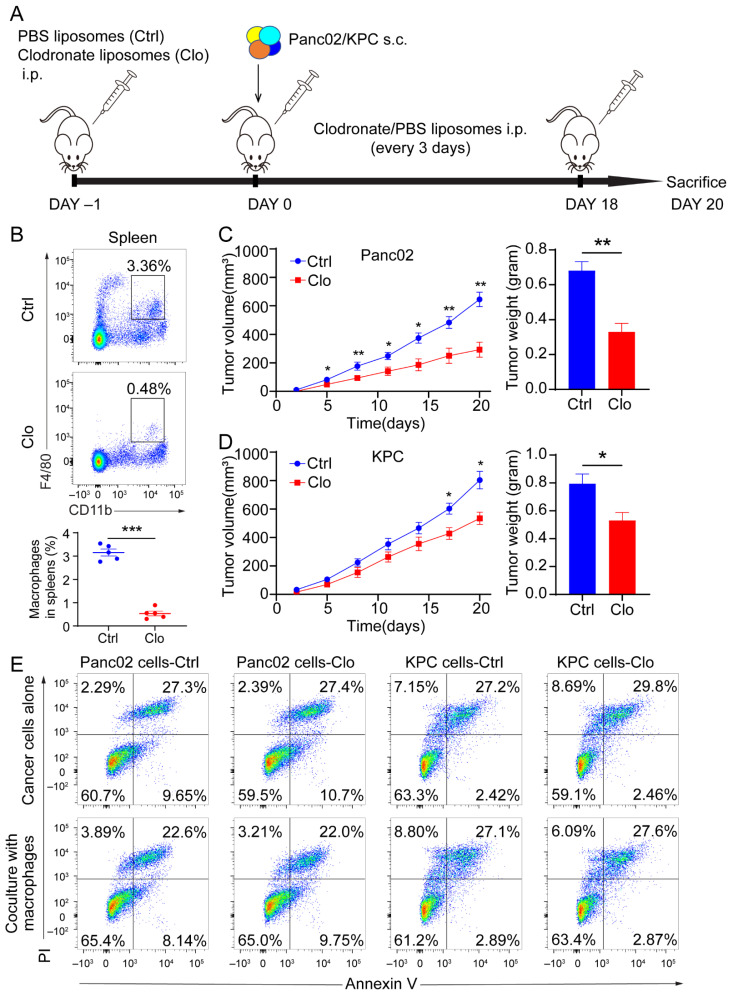
Macrophages exerted a protumor effect in vivo. (**A**) Schematic diagram of our strategy used to target macrophages. C57BL/6 mice were implanted subcutaneously (s.c.) with a KPC-derived tumor cell line or Panc02 cells on day 0, and the mice were treated with either control liposomes or clodronate liposomes at the indicated times by intraperitoneal injection (each 200 µL) (*n* = 7 per group). (**B**) Immune cells isolated from splenocytes were stained with antibodies for the markers CD45, CD11b, and F4/80 and assessed by flow cytometry analysis. The frequency of macrophages (CD11b^+^F4/80^+^) is expressed as a percentage of the gated CD45^+^ cell population in each plot. (**C**,**D**) Left: tumor growth curves showing the volumes of subcutaneously implanted tumors in mice (*n* = 7 per group) treated with control or clodronate liposomes. Right: final tumor weights. (**E**) Top: necrosis and apoptosis of Panc02 and KPC cells after directly treated Panc02 and KPC cells with or without clodronate liposomes determined using Annexin V and PI FACS analysis. Bottom: coculture Panc02 or KPC with macrophages isolated from murine tumor tissues, treated with liposomes. Data are shown as the mean ± SEM. Statistical analyses were carried out by Student’s *t* test. *: *p* ≤ 0.05, **: *p* ≤ 0.01, ***: *p* ≤ 0.001.

**Figure 3 cancers-14-01474-f003:**
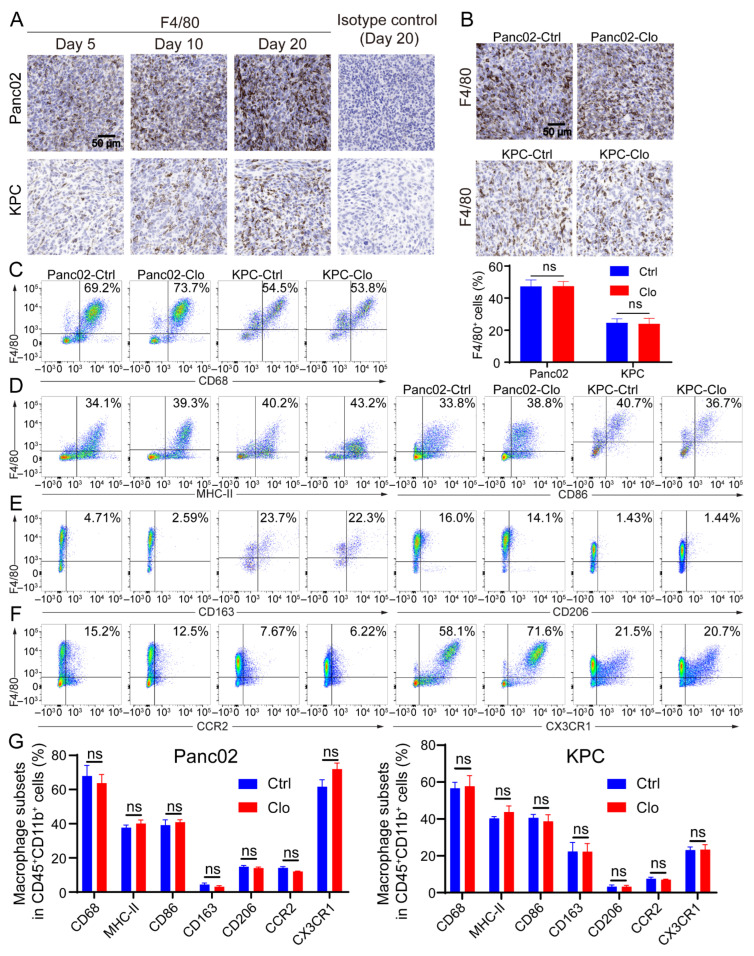
Macrophage populations express a mixture of subset markers with or without macrophage targeting. (**A**) F4/80 (a marker of macrophages) increased as tumor lesions progressed to palpable tumors. Representative IHC images on the 5th, 10th, and 20th days after tumor inoculation and the isotype control (using the slide from day 20) are shown. (**B**) Representative images of pancreatic tissue from the Panc02 and KPC mouse models show a high level of F4/80^+^ macrophage infiltration with or without clodronate liposomes treatment. (**C**–**F**) Cells were isolated from tumors in mice treated with or without clodronate liposomes and analyzed by flow cytometry. For all dot plots, live cells were selected by gating based on zombie staining after gating on CD45^+^ cells, which excluded epithelial cells. CD11b^+^ subpopulations were selected and further analyzed: (**C**) the macrophage markers CD68 and F4/80; (**D**) the M1-like markers MHC-II and CD86; (**E**) the M2-like markers CD163 and CD206; (**F**) the additional markers CCR2 and CX3CR1. (**G**) Statistical diagrams of macrophage subpopulations on live CD45^+^CD11b^+^ myeloid cells are shown. Data are shown as the mean ± SEM. Statistical analyses were carried out by Student’s *t* test. ns: not significant (*p* > 0.05).

**Figure 4 cancers-14-01474-f004:**
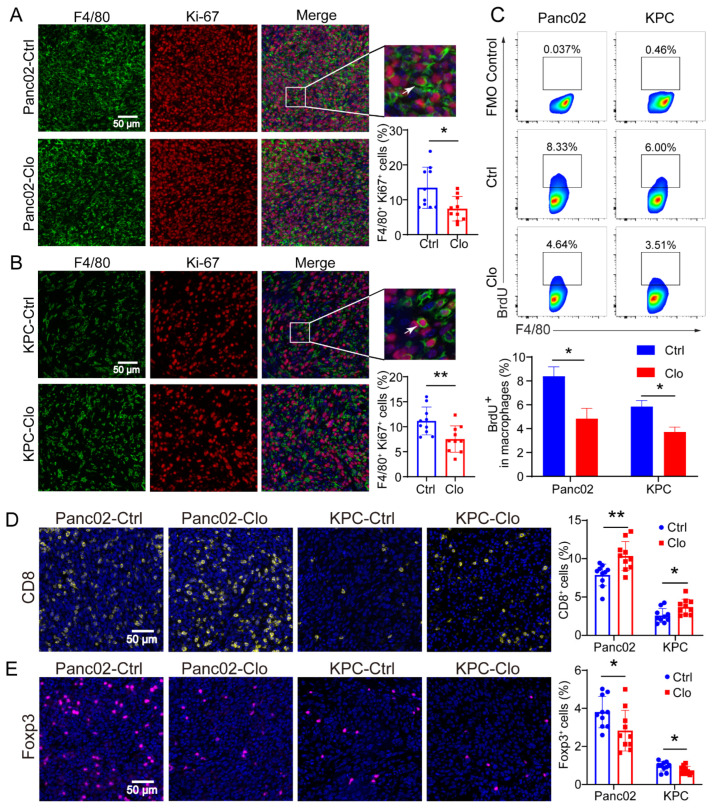
Proliferating F4/80^+^ macrophages were reduced in Panc02 and KPC mouse models after clodronate liposomes treatment. (**A**,**B**) Representative immunofluorescence images of Ki-67 and F4/80 staining in Panc02 or KPC tumors and quantitation of Ki-67 and F4/80 immunofluorescence staining. (**C**) Analysis of BrdU^+^ macrophages in Panc02 and KPC tumor tissues. The animals were injected with BrdU at 12 h and 24 h before sacrifice. (**D**,**E**) Staining of CD8^+^ T and Foxp3^+^ T cells in Panc02 or KPC tumors with or without clodronate liposomes treatment. Data are shown as the mean ± SEM. Statistical analyses were carried out by Student’s *t* test. *: *p* ≤ 0.05, **: *p* ≤ 0.01.

**Figure 5 cancers-14-01474-f005:**
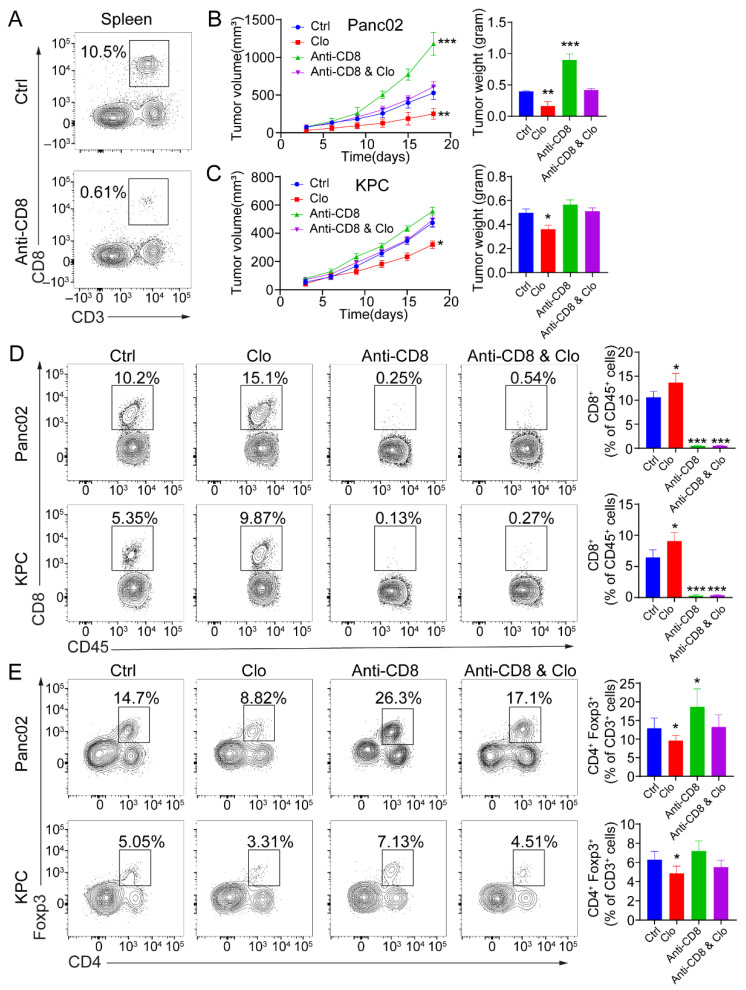
The antitumor response induced by targeting macrophages is CD8^+^ T-cell-dependent. (**A**) Representative flow cytometric analysis of CD8^+^ T cells to confirm depletion was carried out. (**B**,**C**) Mice were treated with 200 μL (intraperitoneal injection) of control liposomes or clodronate liposomes on days −1 and 0 and every 3 days afterward, and/or with isotype control or anti-CD8 antibodies on days −1, 0, 5, 10, and 15 (*n* = 5 mice/group). Left: tumor growth curves showing the volumes of subcutaneously implanted tumors in mice (*n* = 5 per group) treated with control liposomes or clodronate liposomes. Right: final tumor weights. (**D**,**E**) Flow cytometric analysis of tumor-infiltrating CD8^+^ T cells (**D**) and Foxp3^+^ Tregs (**E**). Each data point represents a mouse. Data are shown as the mean ± SEM. Statistical analyses were performed by one-way ANOVA with Tukey’s correction for multiple comparisons. *: *p* ≤ 0.05, **: *p* ≤ 0.01, ***: *p* ≤ 0.001.

**Figure 6 cancers-14-01474-f006:**
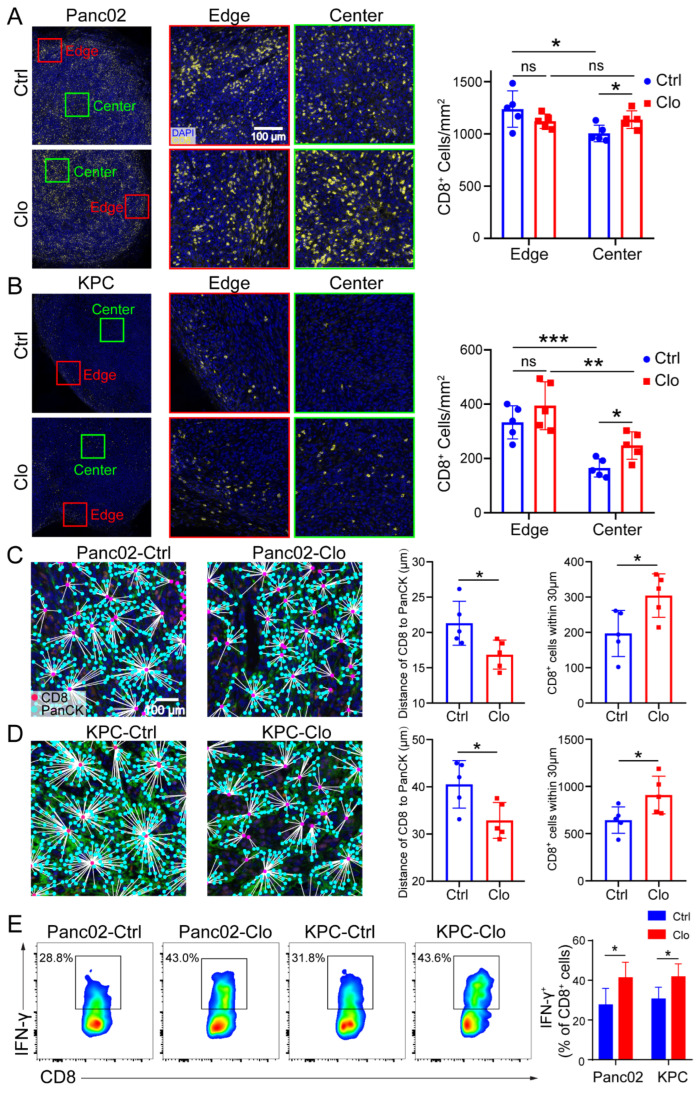
Targeting macrophages promoted CD8^+^ T cell spatial redistribution in tumors. Panc02 and KPC murine pancreatic tumor cells were subcutaneously injected into C57BL/6 mice, and the effects of treatment with control liposomes or clodronate liposomes on intratumoral T cell infiltration and spatial redistribution after macrophage targeting were evaluated. (**A**,**B**) Representative images of the edge (periphery) and center (core) of the tumor sections after anti-CD8 immunostaining are shown. For each tumor, the numbers of CD8^+^ T cells at the edge of the tumor (≤500 µm from the edge of the tumor) and the tumor center (>500 μm from the edge of the tumor) were quantified. The distance between the nuclear centers of a PanCK^+^ tumor cells and their nearest CD8^+^ T cells was calculated for each PanCK^+^ cell identified within the tumor center. (**C**,**D**) The average distances of CD8^+^ T cells to tumor cells in all tumor samples were calculated. (**C**,**D**) (**right**) The distribution of CD8^+^ T cells within 30 μm from tumor cells. Each group contained five mice, corresponding to the five points in the bar graph. We analyzed at least five views for each point and calculated the average density of cells at the center and edge to represent each slide. (**E**) Representative flow cytometric analysis of IFN-γ^+^CD8^+^ cells in CD8^+^ T cells and a statistical graph are shown. Data are shown as the mean ± SEM. Statistical analyses were carried out by Student’s *t* test. *: *p* ≤ 0.05, **: *p* ≤ 0.01, ***: *p* ≤ 0.001, ns: not significant (*p* > 0.05).

## Data Availability

The authors declare that the main data supporting the findings of this study are available within the article and its Appendix A. The data presented in this study are available on request from the corresponding author.

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
