# Peer review of "Targeting Proliferating Tumor-Infiltrating Macrophages Facilitates Spatial Redistribution of CD8+ T Cells in Pancreatic Cancer"

_cancers, 2022, doi:10.3390/cancers14061474_

Round 1
Reviewer 1 Report
The study of Yang et al. aims at characterizing the roles of TAMs in pancreatic cancer. The authors have used clodronate liposomes to deplete macrophages and have observed a higher CD8+ T cells infiltration and a decrease Treg infiltration, leading to reduced tumor volumes in mice. The study brings novel insights about TAM in pancreatic cancer.
Below are my comments:
Figure 1C: have you checked the colocalization of CD68, CD86 and CD163? It looks like images have been taken on different tissue sections.
Figure 2E: you have performed a co-culture of macrophages with cancer cells to study the direct impact of clodronate liposomes on cancer cells viability. Why not studying the impact of clodronate on cancer cells alone, without macrophages? By doing a coculture, you cannot affirm that you study the direct effect of clodronate on cancer cells.
Figure 3A: do the clodronate liposomes affect the resident macrophages of the adjacent normal tissue?
Between 60 to 70% of macrophages are CX3CR1+ in PDAC tumors, which may correspond to recruited monocytes, while only 20% are CX3CR1+ in KPC tumors. The remaining population corresponds to resident macrophages? If you think that clodronate liposomes do not affect recruited macrophages, you should see differences between PDAC and KPC.
Figures 3C – 3F: around 40% of macrophages express MHC II and CD86 pro-inflammatory markers in both PDAC and KPS tumors. Is it known that in pancreatic cancer, macrophages adopt a pro-inflammatory phenotype? In many tumors, TAMs are rather anti-inflammatory.
Figure 4A and B: you said that Ki-67+/F4/80+ population was dramatically reduced. The term is a bit strong since it only represents 3 to 4% of macrophages.
Figure 4E: you did not explain why you see an increased CD8 infiltration and a decrease Treg infiltration when macrophages proliferation is reduced. It would be interesting to study the factors which are released by these macrophages especially since the activation profile of these macrophages did not change (Fig3).
Figure 5C: how can you explain that anti-CD8+clo did not increase the tumor volume at the same level as anti-CD8 only?
Figure 6: Based on the images of KPC tumor, there are no difference of CD8+ localization between the edge and center. You should choose more appropriate images reflecting the quantification results. Moreover, have you checked the enzymatic activity of CD8, based on granzyme and perforin expression?
Author Response
Reviewer1
The study of Yang et al. aims at characterizing the roles of TAMs in pancreatic cancer. The authors have used clodronate liposomes to deplete macrophages and have observed a higher CD8+ T cells infiltration and a decrease Treg infiltration, leading to reduced tumor volumes in mice. The study brings novel insights about TAM in pancreatic cancer.
Below are my comments:
1) Figure 1C: have you checked the colocalization of CD68, CD86 and CD163? It looks like images have been taken on different tissue sections.
Response: Thank you for this suggestion. We detected these markers in series of sections, so they are not identical, but the images presented are from the same field of view. Because colocalization was not the focus of our experiments, we did not perform colocalization analysis of CD68, CD86 and CD163 in one slide.
2) Figure 2E: you have performed a co-culture of macrophages with cancer cells to study the direct impact of clodronate liposomes on cancer cells viability. Why not studying the impact of clodronate on cancer cells alone, without macrophages? By doing a coculture, you cannot affirm that you study the direct effect of clodronate on cancer cells.
Response: Thank you for this comment. Upon adding clodronate liposomes to the cancer cells, we did not find that cell death of the cancer cells was induced; the data have been inserted into Fig. 2E.
3 Figure 3A: do the clodronate liposomes affect the resident macrophages of the adjacent normal tissue?
Between 60 to 70% of macrophages are CX3CR1+ in PDAC tumors, which may correspond to recruited monocytes, while only 20% are CX3CR1+ in KPC tumors. The remaining population corresponds to resident macrophages? If you think that clodronate liposomes do not affect recruited macrophages, you should see differences between PDAC and KPC.
Response: C57BL/6 mice were subcutaneously (s.c.) implanted with a KPC-derived tumor cell line or Panc02 cells, and once tumors had formed, we could not see much of the adjacent normal tissue around the tumor.
At the current cognitive level, monocytes in circulating blood express CCR2, and the original macrophages are CCR2-negative and originate from early embryonic development. Since we did not combine CCR2 and CX3CR1 for analysis, the CXCR3+ population may also contain the CCR2+ population. We could not easily use this marker to define resident macrophages. We did not find that the obvious subpopulation of TAMs was altered by clodronate liposomes, except for proliferating F4/80+ TAMs. However, interestingly, clodronate liposome treatment had a less pronounced anti-tumor effect in KPC tumors than in Panc02 tumors, which may be due to less CD206 M2 expression and a low number of CX3CR1-expressing TAMs (left side of Fig. 3E, 3F). Some subpopulations of TAMs may produce CCL2, CXCL9, and CXCL10, suggesting CD8 T cell trafficking to tumors (PMID: 26109379). The data in Fig. 5 B further support the notion of antitumor immunity through CD8 T cell recruitment and combination with clodronate liposomes treatment in our Panc02 tumor model. We have added this to the Discussion.
4))Figures 3C – 3F: around 40% of macrophages express MHC II and CD86 pro-inflammatory markers in both PDAC and KPS tumors. Is it known that in pancreatic cancer, macrophages adopt a pro-inflammatory phenotype? In many tumors, TAMs are rather anti-inflammatory.
Response: Thank you for the comment. Infiltrating macrophages are dynamic and exhibit enormous plasticity. At early stage, tumor lesions represent a higher expression of immune stimulatory genes (MHC-II etc.) resembling a proinflammatory M1-like phenotype; when progressing to advanced invasive cancer, the expression of immunosuppressive gene (CD206 etc.) is significantly upregulated, becoming the M2-like phenotype (PMID: 29789416). We have added this to the Discussion.
5))Figure 4A and B: you said that Ki-67+/F4/80+ population was dramatically reduced. The term is a bit strong since it only represents 3 to 4% of macrophages.
Response: Thank you for this suggestion. We changed the word "dramatically" to "significantly".
6)) Figure 4E: you did not explain why you see an increased CD8 infiltration and a decrease Treg infiltration when macrophages proliferation is reduced. It would be interesting to study the factors which are released by these macrophages especially since the activation profile of these macrophages did not change (Fig3).
Response: Thank you for the helpful comment. We added a discussion to the revised manuscript. Some cytokines/chemokines, such as CCL22, CCL28, CXCL12, CCL5 and CCL1, can be produced by macrophages (PMID: 30467506). This leads Tregs to be attracted to the tumor sites, which can increase Treg infiltration and decrease CD8+ cell infiltration (PMID: 30705439). Therefore, it would be interesting to study some of the cytokines/chemokines that are released by the subpopulation of macrophages and attract the migration of Tregs, leading to Treg immunosuppression.
7))Figure 5C: how can you explain that anti-CD8+clo did not increase the tumor volume at the same level as anti-CD8 only?
Response: Since anti-CD8 (shown in Fig. 5C) did not significantly increase the tumor volume, we would like to ask if this comment refers to Fig. 5B instead. If so, we consider that anti-CD8+clodronate liposomes did not increase the tumor volume to the degree as anti-CD8 alone, which may be due to effect of Tregs. Tregs are strong immunosuppressive immune cells that play an important role in PDAC progression and are highly expressed in the Panc02 model. Clodronate liposome treatment not only affected the CD8 level in tumors but also promoted a reduction in Tregs. Therefore, the difference in Treg levels between the anti-CD8 and anti-CD8+clodronate liposome groups may be the main reason for the difference in tumor volume. As shown in Fig. 5C, we only observed a difference in the clodronate liposome-treated group, regardless of CD8 depletion.
8))Figure 6: Based on the images of KPC tumor, there are no difference of CD8+ localization between the edge and center. You should choose more appropriate images reflecting the quantification results. Moreover, have you checked the enzymatic activity of CD8, based on granzyme and perforin expression?
Response: We appreciate this advice. To avoid misleading the reader regarding the difference in CD8+ localization, in addition to comparing groups with or without
clodronate liposome treatment, we also performed statistical analysis between the edge and the center group, and the data are shown in Fig. 6A and 6B.
We checked the enzymatic activity of CD8 based on granzyme B and found no difference between groups; the data have been added to Fig. S4.

Reviewer 2 Report
The manuscript by Yang et al entitled “Targeting Proliferating Tumor-Infiltrating Macrophages Facilitates Spatial Redistribution of CD8+ T Cells in Pancreatic Cancer” explored how clodronate liposomes regulates CD8+ T cell spatial redistribution in PDAC by regulating macrophages. Overall, although this study looks interesting and provides some new findings, there are several drawbacks that dampen the enthusiasm. The following concerns need to be addressed.
- As shown in Fig. 2B, intra-peritoneal injection of clodronate liposomes can significantly decrease the abundance of CD11b+F4/80+ macrophages. However, in Fig. 3B, the authors showed that treatment with clodronate liposomes cannot reduce the number of macrophages in the tumor microenvironment. Did they mean these are the resident macrophages, rather than those derived from monocytes? They explained in the discussion that “loss of TAMs can be compensated by recruitment of circulating Ly6Ghi monocytes.”. They need to provide evidence to show that clodronate liposomes did not decrease the abundance of monocytes.
- They examined the abundance of CD11b+F4/80+ macrophages in the spleen. Did they examine the abundance of CD11b+ cells in the spleen and the tumor microenvironment?
- Please double check Fig. 3E and 3F. I am concerned about the gating of the flow cytometry. Usually we set >103 as positive.
- In the result section 3.3, the authors stated that “Macrophage and tumor cell crosstalk via CX3CR1 and CCR2 could be the mechanism driving PDAC.”. But they don’t have further evidence to support this claim and this manuscript is not focusing on this hypothesis. Please clarify.
- They stated that “Figure 4. F4/80+ macrophages proliferation was inhibited in Panc02 and KPC mouse models.”. This sentence is misleading. Please rephrase it.
- Although they showed in Fig. 4 that clodronate liposomes can decrease the proliferating F4/80+ TAMs, the phenotypes or the subpopulations of macrophages remained the same. In other words, the number of M2 TAMs did not decrease more dramatically than the M1 TAMs. How would it regulate CD8+ T cell spatial redistribution in tumors? What’s the mechanism?
- How can they demonstrate clodronate liposomes regulated CD8+ T cell spatial redistribution in tumors is dependent on the regulation of the proliferating F4/80+ TAMs? What if clodronate liposomes can directly regulate Tregs and CD8+ T cell spatial redistribution?
- What did they mean by “Here, our study found that after PDAC-bearing mice were treated with clodronate liposomes, reduced numbers of BrdU incorporated and Ki-67+ proliferating macrophages, which might be maintained by local proliferation.” Please find a native English speaker to correct the grammar mistakes and polish the manuscript.
Author Response
Reviewer2
The manuscript by Yang et al entitled “Targeting Proliferating Tumor-Infiltrating Macrophages Facilitates Spatial Redistribution of CD8+ T Cells in Pancreatic Cancer” explored how clodronate liposomes regulates CD8+ T cell spatial redistribution in PDAC by regulating macrophages. Overall, although this study looks interesting and provides some new findings, there are several drawbacks that dampen the enthusiasm. The following concerns need to be addressed.
1 As shown in Fig. 2B, intra-peritoneal injection of clodronate liposomes can significantly decrease the abundance of CD11b+F4/80+ macrophages. However, in Fig. 3B, the authors showed that treatment with clodronate liposomes cannot reduce the number of macrophages in the tumor microenvironment. Did they mean these are the resident macrophages, rather than those derived from monocytes? They explained in the discussion that “loss of TAMs can be compensated by recruitment of circulating Ly6Ghi monocytes.”. They need to provide evidence to show that clodronate liposomes did not decrease the abundance of monocytes.
Response: We have modified this sentence phrase to, “we speculated that the loss of TAMs can be compensated by the recruitment of circulating Ly6Ghi monocytes”, which comes from the evidence provided in the Zhu 2017 Immunity paper. This article reports the treatment of PDAC tumor-naive mice with CSF1-neutralizing antibodies in combination with clodronate liposomes, followed by a 10-day chase period to allow circulating monocyte numbers in the mice to recover. The authors found that circulating monocyte numbers were restored to control levels. Pancreatic-resident macrophages also recovery to certainly levels unless macrophages were depleted during implantation. This finding indicates that tissue-resident macrophages in PDAC might originate from embryonic hematopoiesis (PMID: 28813661).
2 They examined the abundance of CD11b+F4/80+ macrophages in the spleen. Did they examine the abundance of CD11b+ cells in the spleen and the tumor microenvironment?
Response: We appreciate this advice. We examined the abundance of CD11b+ cells in the spleen and the tumor microenvironment, and the data have been added to Fig. S2.
Selective depletion of spleen cell populations after treatment with clodronate liposomes
showed significant depletion of F4/80+ and without pronounced depletion of CD11b+ populations.
3 Please double check Fig. 3E and 3F. I am concerned about the gating of the flow cytometry. Usually we set >103 as positive.
Response: Thank you for this suggestion. We set the gate according to the FMO control, and the specific gating strategy has been added to Fig. S3.
4) In the result section 3.3, the authors stated that “Macrophage and tumor cell crosstalk via CX3CR1 and CCR2 could be the mechanism driving PDAC.”. But they don’t have further evidence to support this claim and this manuscript is not focusing on this hypothesis. Please clarify.
They stated that “Figure 4. F4/80+ macrophages proliferation was inhibited in Panc02 and KPC mouse models.”. This sentence is misleading. Please rephrase it.
Response: Thank you for the suggestion. We deleted the sentence, “Macrophage and tumor cell crosstalk via CX3CR1 and CCR2 could be the mechanism driving PDAC,” and we have rephrased the sentence, “Figure 4. F4/80+ macrophages proliferation was inhibited in Panc02 and KPC mouse models,” to “Figure 4. Proliferating F4/80+ macrophages were reduced in Panc02 and KPC mouse models after clodronate liposomes treatment.”
5) Although they showed in Fig. 4 that clodronate liposomes can decrease the proliferating F4/80+ TAMs, the phenotypes or the subpopulations of macrophages remained the same. In other words, the number of M2 TAMs did not decrease more dramatically than the M1 TAMs. How would it regulate CD8+ T cell spatial redistribution in tumors? What’s the mechanism?
Response: Thank you for this helpful comment. We added a related discussion to the revised manuscript. TAMs exhibit enormous plasticity and can exhibit M1/M2 expression depending on tumor stage or local microenvironmental cues. We did not find that the obvious subpopulation of TAMs was altered by clodronate liposomes, except for proliferating F4/80+ TAMs. However, interestingly, clodronate liposomes treatment had a less pronounced antitumor effect in KPC tumors than in Panc02 tumors, which may be due to less CD206 M2 expression and the low number of CX3CR1-expressing TAMs (left side of Fig. 3E, 3F). Some subpopulations of TAMs may produce CCL2, CXCL9, and CXCL10, suggesting CD8 T cell trafficking to tumors (PMID: 26109379). The data in Fig. 5B further support the notion of antitumor immunity through CD8 T cell recruitment. In the future, which subpopulation of TAMs is affected by clodronate liposomes still needs to be further investigated.
6) How can they demonstrate clodronate liposomes regulated CD8+ T cell spatial redistribution in tumors is dependent on the regulation of the proliferating F4/80+ TAMs? What if clodronate liposomes can directly regulate Tregs and CD8+ T cell spatial redistribution?
Response: Thank you for this helpful comment. Some cytokines/chemokines, such as CCL22, CCL28, CXCL12, CCL5 and CCL1, can be produced by macrophages (PMID: 30467506). This leads Tregs to be attracted to tumor sites, which can increase Treg infiltration and decrease CD8 infiltration (PMID: 30705439). Therefore, it would be interesting to study some of the cytokines/chemokines that are released by the subpopulation of macrophages and attract the migration of Tregs, leading to Treg immunosuppression.
The mechanism by which clodronate liposomes deplete macrophages is based on the phagocytosis of macrophages. In detail, macrophages phagocytose clodronate-containing liposomes, followed by the gradual release of clodronate into the cell by lysosomal phospholipases and the subsequent induction of macrophage apoptosis. This specific mechanism did not apply to Tregs and CD8+ T cells.
7) What did they mean by “Here, our study found that after PDAC-bearing mice were treated with clodronate liposomes, reduced numbers of BrdU incorporated and Ki-67+ proliferating macrophages, which might be maintained by local proliferation.” Please find a native English speaker to correct the grammar mistakes and polish the manuscript.
Response: Thank you for the suggestion. We have corrected this sentence to, “Here, we
found that after pancreatic ductal adenocarcinoma (PDAC)-bearing mice were treated with clodronate liposomes, the numbers of BrdU-incorporated and Ki-67+ proliferating macrophages were reduced, which might be maintained by local proliferation.” The language in the manuscript has been polished by an English language editing company (American Journal Experts), and the manuscript is accompanied by an editing certificate. We have also carefully checked the entire manuscript to correct typographical and grammatical errors. Please see our corrections in the revised manuscript.

Reviewer 3 Report
In this study, the authors exploited clodronate liposome to treat the mice subcutaneously inoculated with Panc 02 or KPC-derived pancreatic ductal adenocarcinoma (PDAC) cells. They found that clodronate liposome repressed the proliferation of tumor-infiltrating macrophages (TIMs) without alteration of their M1 or M2-polarization. Furthermore, they noticed that targeting proliferating TIMs can increase the tumor-infiltrating level of cytotoxic T cells [i.e. CD8(+)IFN-r(+) T cells]. This report is interesting, but the following comments need to be addressed before the submission can be finally accepted for publication.
(1). In the 4th line of the Abstract section, the use of “impaired” needs to be corrected.
(2). There are many questionable points in the 4th paragraph of the Introduction section, which need to be addressed by extensive modification or even deletion. First, some statements are incomplete or inadequate. The authors stated “the high degree of TAM infiltration is closely related to the poor prognosis of most types of cancer [18], such as glioblastoma [19], bladder cancer [20] and PDAC [21]”, but actually in PDAC as example, M2-polarized TAMs but not general TAMs are correlated with metastasis and poorer prognosis of PDAC patients [Ref: J Surg Res (2011) 167: e211–219 and Oncoimmunology (2018) 7: e1424612]. Additionally, the authors stated “impeding tissue resident macrophage formation is vital in obstructing PDAC development, improving an- titumor immunity, and clinical treatment”, but actually it is well known that the protumor role of myeloid-derived tumor-infiltrating macrophages is not inferior to tissue-resident macrophages at all. Many sentences exhibit grammar errors and unreadable, for example, “TAMs promote progression on the Warburg effect…” and “Ideal is to defining subset…”.
(3). In the last paragraph of the Introduction section, the statement “The results demonstrated that targeting proliferating F4/80+ macrophages in tumor tissue and diminished Foxp3+ regulatory T cell (Treg)-mediated inhibition of the antitumor activity of infused effector CD8+ T cells” needs to be modified to be readable.
(4). Please indicate the case numbers analyzed in Figure 1A &1B. If based on the statement of Figure S1 legend, only 90 of 178 cases of TCGA-PAAD dataset were analyzed in the Figures. Why did the study select samples? Because there have been reports showing higher infiltrating levels of M2-polarized macrophages rather than pan-macrophages are significantly correlated with metastasis and poorer prognosis of PDAC patients [Ref: J Surg Res (2011) 167: e211–219 and Oncoimmunology (2018) 7: e1424612]. The authors should perform the statistical analyses of 178 PDAC samples but not only 90 of them.
(5). The tumorigeneity of Panc 02 cells is not comparable to KPC-derived cells. However, the questionable data of Figure 2C shows that it only took 5 days for 500000 Panc 02 cells to grow up to a 100-mm3 tumor mass, which is almost as well as that exhibited by KPC-derived cells.
(6). The text describing the experimental condition of Figure 2E is not clearly presented so as we cannot judge whether the interpretation of Figure 2E data is adequate or not. It is unclear whether the same amount of macrophages isolated from control and clodronate-treated mice, respectively, were used for coculture with KPC-derived cells or Panc 02 cells. Moreover, how to know the died cells from cancer cells or from macrophages?
(7). In Figure 3B legend, please clarify the images standing for pancreatic tissues or tumor tissues.
(8). Based on the data of Figure 4-6, the authors conclude that targeting proliferating F4/80+ TAMs can increase the tumor-infiltrating level of cytotoxic T cells [i.e. CD8(+)IFN-r(+) T cells]. Why did not the authors also detect the levels of effective T cells [i.e. CD4(+)IFN-r(+) T cells]? Additionally, the authors should interpret why these cytotoxic T cells did not cause cancer cell death when cocultivating with them (Figure 2E).
Author Response
Reviewer 3
In this study, the authors exploited clodronate liposome to treat the mice subcutaneously inoculated with Panc 02 or KPC-derived pancreatic ductal adenocarcinoma (PDAC) cells. They found that clodronate liposome repressed the proliferation of tumor-infiltrating macrophages (TIMs) without alteration of their M1 or M2-polarization. Furthermore, they noticed that targeting proliferating TIMs can increase the tumor-infiltrating level of cytotoxic T cells [i.e. CD8(+)IFN-r(+) T cells]. This report is interesting, but the following comments need to be addressed before the submission can be finally accepted for publication.
(1). In the 4th line of the Abstract section, the use of “impaired” needs to be corrected.
Response: Thank you for this suggestion. We changed this word to “impeded”.
(2). There are many questionable points in the 4th paragraph of the Introduction section, which need to be addressed by extensive modification or even deletion. First, some statements are incomplete or inadequate. The authors stated “the high degree of TAM infiltration is closely related to the poor prognosis of most types of cancer [18], such as glioblastoma [19], bladder cancer [20] and PDAC [21]”, but actually in PDAC as example, M2-polarized TAMs but not general TAMs are correlated with metastasis and poorer prognosis of PDAC patients [Ref: J Surg Res (2011) 167: e211–219 and Oncoimmunology (2018) 7: e1424612]. Additionally, the authors stated “impeding tissue resident macrophage formation is vital in obstructing PDAC development, improving an- titumor immunity, and clinical treatment”, but actually it is well known that the protumor role of myeloid-derived tumor-infiltrating macrophages is not inferior to tissue-resident macrophages at all. Many sentences exhibit grammar errors and unreadable, for example, “TAMs promote progression on the Warburg effect…” and “Ideal is to defining subset…”.
Response: We appreciate this advice. We modified and deleted some sentences in this paragraph. The sentences of concern have been adjusted to the following:
“Preclinical and clinical data indicate that a high degree of TAM infiltration is related to the poor prognosis of some types of cancer [18], such as glioblastoma [19] and bladder cancer [20]. On the other hand, in some cancers, such as colorectal cancer [21] and ovarian cancer [22], TAM infiltration is associated with a good prognosis. The difference between these results can be attributed to not only different cancer types but also some intratumor factors, such as the TAM distributions in the different TMEs. Elevated levels of TAMs in the tumor stroma are associated with poor prognosis in non-small-cell lung cancer (NSCLC), and the degree to which TAM penetrate cancerous islets is associated with good prognosis [23,24]. These findings demonstrate the inter- and intratumoral heterogeneity of TAMs, which may be related to the ontogeny and location of TAMs in the TME. The activation status of TAMs, such as M1 or M2 polarization, also contributes to cancer progression; for example, in PDAC, M2-polarized TAMs, rather than general TAMs, are correlated with metastasis and poorer prognosis in PDAC patients [3,25].”
“Once we redefine the subsets of TAMs, we can properly target these myeloid-derived tumor-infiltrating macrophages, halt PDAC tumor progression, and improve antitumor immunity and clinical outcomes.”
“TAMs promote PDAC progression by modulating the Warburg effect through the CCL18/NF-κB/VCAM-1 pathway.”
“The ideal approach would be to define the subset of TAMs that can contribute to PDAC development and to carry out therapeutic intervention.”
(3). In the last paragraph of the Introduction section, the statement “The results demonstrated that targeting proliferating F4/80+ macrophages in tumor tissue and diminished Foxp3+ regulatory T cell (Treg)-mediated inhibition of the antitumor activity of infused effector CD8+ T cells” needs to be modified to be readable.
Response: Thank you for this suggestion. We modified this sentence to, “The results demonstrated that targeting proliferating F4/80+ macrophages might alter the myeloid population and is associated with reduced Foxp3+ Tregs and increased CD8+ T cell infiltration. Targeting proliferating macrophages also enhanced the antitumor effect of CD8+ T lymphocytes and promoted CD8+ T cell spatial redistribution in tumors, contributing to a more antitumor immune environment.”
(4). Please indicate the case numbers analyzed in Figure 1A &1B. If based on the statement of Figure S1 legend, only 90 of 178 cases of TCGA-PAAD dataset were analyzed in the Figures. Why did the study select samples? Because there have been reports showing higher infiltrating levels of M2-polarized macrophages rather than pan-macrophages are significantly correlated with metastasis and poorer prognosis of PDAC patients [Ref: J Surg Res (2011) 167: e211–219 and Oncoimmunology (2018) 7: e1424612]. The authors should perform the statistical analyses of 178 PDAC samples but not only 90 of them.
Response: Thank you for this suggestion. We have indicated the case numbers analyzed in Fig. 1A and 1B. In Fig. S1, we used a total of 179 cases from the GEPIA dataset, and the group cutoff was set as the quartile (cutoff-high (%) was 75%, and cutoff-low (%) was 25%). To avoid misleading the reader, we have modified the description in the Materials and Methods to “For overall survival analysis based on CD68, CD86 and CD163 expression, 179 patients were divided into either a high expression group or a low expression group using the quartile value of TPM expression as the cutoff value based on the GEPIA data,” and adjusted the Fig. S1A legend to “Overall survival curve analysis based on CD68, CD86 and CD163 TPM expression in 179 pancreatic tumor cases from the GEPIA database. The “High” and “Low” group cutoff was set as the quartile (cutoff-high (%) was 75%, and cutoff-low (%) was 25%). n (high)=45, n (low)=45,” in the revised submission.
(5). The tumorigeneity of Panc 02 cells is not comparable to KPC-derived cells. However, the questionable data of Figure 2C shows that it only took 5 days for 500000 Panc 02 cells to grow up to a 100-mm3 tumor mass, which is almost as well as that exhibited by KPC-derived cells.
Response: We appreciate this comment. The difference in tumorigenicity between the Panc02 and KPC models is dependent on the aggressiveness of tumor cells or the number of injected cells at the starting point. We omitted the number of injected KPC cells; we have added this number (3×105) to the revised manuscript.
(6). The text describing the experimental condition of Figure 2E is not clearly presented so as we cannot judge whether the interpretation of Figure 2E data is adequate or not.
It is unclear whether the same amount of macrophages isolated from control and clodronate-treated mice, respectively, were used for coculture with KPC-derived cells or Panc 02 cells. Moreover, how to know the died cells from cancer cells or from macrophages?
Response: Thank you for this suggestion. We added a detailed description of the coculture experiment to the Materials and Methods section: “For coculture analysis, TAMs and the Panc02 or KPC cell line were cocultured using a noncontact coculture Transwell system (Corning, NY, USA). TAMs isolated from tumor tissues were seeded in 0.4 μμm pores (1 × 105 cells per pore), and 6-well plates were seeded with Panc02 or KPC cancer cells (1 × 105 cells per well). We administered clodronate liposomes or control liposomes to the cancer cells. After 48 hours of coculture, the Panc02 and KPC cells were harvested for further analyses.” In addition, we modified Fig. 2E by inserting the data showing the direct impact of clodronate liposomes on cancer cells without coculture conditions and found no direct impact.
(7). In Figure 3B legend, please clarify the images standing for pancreatic tissues or tumor tissues.
Response: Thank you for this comment. Fig. 3B shows the tumor tissues. We revised the figure legend and checked the full text.
(8). Based on the data of Figure 4-6, the authors conclude that targeting proliferating F4/80+ TAMs can increase the tumor-infiltrating level of cytotoxic T cells [i.e. CD8(+)IFN-r(+) T cells]. Why did not the authors also detect the levels of effective T cells [i.e. CD4(+)IFN-r(+) T cells]? Additionally, the authors should interpret why these cytotoxic T cells did not cause cancer cell death when cocultivating with them (Figure 2E).
Response: We found that targeting proliferating F4/80+ TAMs could increase the level of tumor-infiltrating cytotoxic CD8 T cells (Fig. 6E) and suppress tumor progression in a CD8 T cell-dependent manner (Fig. 5B). We can exclude the influence of the number of infiltrating CD4+ cells (Fig. 5E, the total CD4+ T cell population remained relatively similar), but we cannot exclude the CD4(+)IFN-r(+) T cells. The future plan is to look for functional cell subsets and populations which exert an immunosuppressive effect in the PDAC microenvironment, in detail, to detect the following: 1) effective and immunosuppressive phenotypes of infiltrating CD4 T cells to see whether there is a switch from Th1 “effector” cells (IFN-rhi, IL-10low, PD-1low) to cells with a suppressive phenotype (IFN-rlow, IL-10high, PD-1high), 2) the functional state of CD8, which might switch from an “effector” phenotype (IFN-rhi, CD107high, PD-1low) to an “exhausted” phenotype (IFN-rlow, CD107low, PD-1high), 3) the expression of PD-L1 in resident macrophages, and 4) the infiltrating Treg cells with a more immunosuppressive phenotype (IL-10high). We have added this to the Discussion.
For the coculture of cytotoxic T cells and tumor cells, although this may require further study, the coculture condition is hardly achievable for it’s difficult for cytotoxic T cells cause cancer cell death in vitro, unlike in tumor tissue where the numerous TILs are responsible for the killing of tumor cells.

Round 2
Reviewer 2 Report
The authors have addressed all my concerns.
Author Response
We checked the grammar and spelling. The quality of the manuscript has been improved by our further modification.
Reviewer 3 Report
The authors should respond again the following comments:
(1). In response to the original comment 1, the authors use “impeded” instead of “impaired”. However, targeting proliferating macrophages described herein should “facilitated” or “provided advantages to” CD8(+) T cell….. .
(2). In Figure 2A legend, the number of PDAC samples is 178, but in Figure 1B the PDAC number is 179. Why is the inconsistence?
(3). The authors should interpret the data of Figures 4-6 in which targeting proliferating F4/80+ TAMs can increase the tumor-infiltrating level of cytotoxic T cells [i.e. CD8(+)IFN-r(+) T cells but why these cytotoxic T cells did not cause cancer cell death.
Author Response
The authors should respond again the following comments:
(1). In response to the original comment 1, the authors use “impeded” instead of “impaired”. However, targeting proliferating macrophages described herein should “facilitated” or “provided advantages to” CD8(+) T cell….. .
Response: We appreciate this comment and changed this word to “facilitated”.
(2). In Figure 2A legend, the number of PDAC samples is 178, but in Figure 1B the PDAC number is 179. Why is the inconsistence?
Response: Thank you for this suggestion. In Figure 1A, we use data from TCGA database, so the number of PDAC samples is 178. As for Figure 1B, we analyzed gene expression in pancreatic cancer tissues and normal tissues, since the TCGA database only contains 4 normal tissues, we used the GEPIA database that contains data from TCGA and GTEx, and ended up with 179 PDAC samples and 171 normal tissue samples. Details are described in Materials and Methods 2.3. For better readability, we added the description in the figure legend.
(3). The authors should interpret the data of Figures 4-6 in which targeting proliferating F4/80+ TAMs can increase the tumor-infiltrating level of cytotoxic T cells [i.e. CD8(+)IFN-r(+) T cells but why these cytotoxic T cells did not cause cancer cell death.
Response: Thank you for this suggestion. We agree with the comment on targeting proliferating F4/80+ TAMs can increase the tumor-infiltrating CD8(+)IFN-r(+) cytotoxic T cells and cause cancer cell death in vivo. But in our Fig2E (in vitro experiments), we cocultured tumor cells with macrophages isolated from murine tumor tissues and we did not expect to see cancer cell death because we didn’t add cytotoxic T cells.
